# The DWORF micropeptide enhances contractility and prevents heart failure in a mouse model of dilated cardiomyopathy

Catherine A Makarewich[1], Amir Z Munir[1], Gabriele G Schiattarella[2], Svetlana Bezprozvannaya[1], Olga N Raguimova[3], Ellen E Cho[3], Alexander H Vidal[1], Seth L Robia[3], Rhonda Bassel-Duby[1], Eric N Olson[1]*

[1]Department of Molecular Biology and Hamon Center for Regenerative Science and Medicine, University of Texas Southwestern Medical Center, Dallas, United States; [2]Department of Internal Medicine, University of Texas Southwestern Medical Center, Dallas, United States; [3]Department of Cell and Molecular Physiology, Loyola University Chicago, Maywood, United States

**Abstract** Calcium ($Ca^{2+}$) dysregulation is a hallmark of heart failure and is characterized by impaired $Ca^{2+}$ sequestration into the sarcoplasmic reticulum (SR) by the SR-$Ca^{2+}$-ATPase (SERCA). We recently discovered a micropeptide named DWORF (DWarf Open Reading Frame) that enhances SERCA activity by displacing phospholamban (PLN), a potent SERCA inhibitor. Here we show that DWORF has a higher apparent binding affinity for SERCA than PLN and that DWORF overexpression mitigates the contractile dysfunction associated with PLN overexpression, substantiating its role as a potent activator of SERCA. Additionally, using a well-characterized mouse model of dilated cardiomyopathy (DCM) due to genetic deletion of the muscle-specific LIM domain protein (MLP), we show that DWORF overexpression restores cardiac function and prevents the pathological remodeling and $Ca^{2+}$ dysregulation classically exhibited by MLP knockout mice. Our results establish DWORF as a potent activator of SERCA within the heart and as an attractive candidate for a heart failure therapeutic.

DOI: https://doi.org/10.7554/eLife.38319.001

*For correspondence:
Eric.Olson@UTSouthwestern.edu

## Introduction

Cardiovascular disease is the leading cause of death and disability in industrialized nations and its prevalence is rising rapidly. The molecular mechanisms that drive the progression of heart failure are poorly understood due to the complex and multifactorial nature of the disease. Among the many pathological features of heart failure, the most prominent and widespread is aberrant $Ca^{2+}$ cycling, which reduces myocardial contractility and initiates pathological remodeling (*Piacentino et al., 2003*). Dilated cardiomyopathy (DCM), which is characterized by ventricular chamber enlargement and systolic dysfunction, is the third most common cause of heart failure and the most frequent reason for heart transplantation (*Maron et al., 2006*). While many cases of DCM are idiopathic in nature, direct links have been established between the development of DCM as a consequence of inflammatory, metabolic, or toxic insults or by genetic mutations in $Ca^{2+}$ regulatory proteins, contractile proteins or cytoskeletal proteins that reside at the sarcomeric Z-disc (*Arber et al., 1997*; *Cahill et al., 2013*; *McNally et al., 2013*). Regardless of the cause of DCM, progressive chamber dilation and heart failure are driven by $Ca^{2+}$ dysregulation including alterations in $Ca^{2+}$ cycling and homeostasis (*Luo and Anderson, 2013*; *Minamisawa et al., 1999*).

$Ca^{2+}$ is a ubiquitous intracellular second messenger involved in the regulation of a broad range of cellular processes including muscle contraction, energy metabolism, proliferation and apoptosis. The

**eLife digest** The heart is a muscular organ that contracts regularly to pump blood around the body, ensuring that nutrients and oxygen are carried to the cells and organs. Heart failure is a disease where the heart muscle becomes weakened, does not beat as strongly, and cannot pump blood as well as it should. Eventually, the heart can no longer deliver enough blood to meet the body's needs.

Although heart failure is a widespread disease, we still do not fully understand its underlying causes and the molecular machinery driving its progression. However, one common feature in many cases of heart failure is a problem with the supply of calcium to the heart muscle. Calcium is the molecule responsible for the process of muscle contraction; the strength of contraction depends on the amount of calcium available. Movement of calcium within heart cells is in turn controlled by an enzyme pump called SERCA.

In 2016, researchers identified a small protein, DWORF, which increased the activity of SERCA. Makarewich et al. – including many of the researchers involved in the 2016 study – therefore wanted to find out more about how DWORF and SERCA worked together. They also wanted to test if DWORF could be used to boost the heart's ability to pump blood efficiently, and if so, whether it could treat heart failure.

Genetically modified mice that produced larger than normal amounts of DWORF had more available calcium in the heart muscle, which made it contract more strongly. This was true even when the same mice were treated with an excessive amount of a specific protein (phospholamban) that can lower the activity of SERCA, suggesting that DWORF might have a protective effect on the heart. Experiments using mice engineered to show symptoms of heart disease confirmed that DWORF treatment did indeed help their hearts beat normally, and, crucially, prevented them from developing heart failure.

This work has shown for the first time that DWORF can restore the heart's ability to pump normally in an experimental model of heart disease. In the future, Makarewich et al. hope that DWORF could be a useful target for new, more effective drugs to treat heart failure.
DOI: https://doi.org/10.7554/eLife.38319.002

involvement of $Ca^{2+}$ in so many fundamental events demands its precise control, which predominantly occurs at the level of the sarco(endo)plasmic reticulum (SR), the major intracellular $Ca^{2+}$ storage site. In the heart, $Ca^{2+}$ plays a crucial role in connecting membrane excitability with contraction, a process known as excitation contraction-coupling. During each cycle of contraction and relaxation, $Ca^{2+}$ is released from the SR via ryanodine receptors (RyRs) into the cytoplasm where it binds to myofilament proteins to induce sarcomere shortening (*Bers, 2002*). Relaxation is initiated by $Ca^{2+}$ re-sequestration into the SR, a process that is mediated by a SR $Ca^{2+}$-ATPase (SERCA), which uses the energy generated from ATP hydrolysis to pump $Ca^{2+}$ against its concentration gradient back into the lumen of the SR.

A universal cause of the decreased contractile performance of the failing heart is impaired $Ca^{2+}$ sequestration into the SR and a reduction in SERCA activity and protein (*Luo and Anderson, 2013*). Hence, augmenting SERCA activity has been suggested as an attractive clinical approach for treating heart failure by preserving cardiac contractile function (*Gwathmey et al., 2013*; *Kranias and Hajjar, 2012*; *Pleger et al., 2013*). Consistent with this hypothesis, overexpression of SERCA2a, the predominant cardiac isoform of SERCA, has been shown to improve cardiac function and ameliorate the progression of cardiovascular disease in several rodent and large animal models of heart failure (*Kawase et al., 2008*; *Miyamoto et al., 2000*; *Prunier et al., 2008*). In the heart, it has been shown that the activity of SERCA is inhibited by the binding of two small transmembrane peptides, phospholamban (PLN) and sarcolipin (SLN), which lower the affinity of SERCA for $Ca^{2+}$ and decrease the rate of $Ca^{2+}$ re-uptake into the SR (*MacLennan and Kranias, 2003*; *Nelson et al., 2014*; *Vangheluwe et al., 2006*). Our lab recently discovered and characterized a novel micropeptide named DWarf Open Reading Frame (DWORF), which binds directly to SERCA and enhances its activity by displacing the SERCA inhibitory peptides PLN and SLN (*Nelson et al., 2016*). The discovery of DWORF as a potent stimulator of SERCA activity and cardiac contractility provides a novel

therapeutic target through which to preserve cardiac contractile function and restore Ca$^{2+}$ homeostasis in the context of heart failure.

In this study, we investigate the molecular determinants of the DWORF-SERCA regulatory complex and explore the therapeutic potential of DWORF overexpression as a means to increase SERCA activity and cardiac contractility in the context of heart failure. We use a combination of techniques to examine the interaction of SERCA with DWORF and PLN to precisely demonstrate that SERCA has a higher apparent affinity for DWORF than for PLN. We also examine the stoichiometric parameters of the DWORF-SERCA complex and analyze the ability of DWORF and PLN to homo- or hetero-oligomerize into higher order structures. Additionally, we show that in vivo overexpression of DWORF relieves the inhibitory effects of PLN on SERCA, even in the context of super-inhibition of SERCA via cardiac-specific PLN overexpression. Lastly, we use a well-characterized mouse model of DCM, muscle-specific LIM protein (MLP) knockout mice, to show that DWORF overexpression enhances cardiac function and prevents adverse cardiac remodeling to abrogate the heart failure phenotype observed in these mice (*Arber et al., 1997*). The MLP KO mouse model of heart failure strongly reproduces the morphological and clinical characteristics of DCM and heart failure in human patients, which highlights the potential of DWORF overexpression as a clinically relevant therapy (*Hoshijima et al., 2006*).

## Results

### SERCA has a higher apparent affinity for DWORF than for PLN and interacts with both proteins in a 1:1 stoichiometry

The interaction of PLN with SERCA has been extensively studied (*Kranias and Hajjar, 2012*; *MacLennan and Kranias, 2003*; *Hou et al., 2008*; *Hou and Robia, 2010*; *Kelly et al., 2008*; *Kimura et al., 1998*; *Robia et al., 2007*). In contrast, due to the very recent discovery of DWORF, very little is known about the SERCA/DWORF regulatory complex. To examine the apparent binding affinity and stoichiometry of SERCA in complex with DWORF in live cell membranes, we performed fluorescence resonance energy transfer (FRET) experiments using transfected AAV-293 cells. We sampled large populations of cells (~1000 cells per experiment) coexpressing mCerulean (Cer)-SERCA2a and either yellow fluorescent protein (YFP)-DWORF or –PLN and compared each cell's FRET efficiency (Cer excitation, YFP emission) with its YFP-DWORF or –PLN fluorescence intensity, which is an index of protein expression (*Hou et al., 2008*; *Kelly et al., 2008*). For both DWORF and PLN, FRET efficiency increased with increasing protein expression, a relationship that can be approximated by a hyperbolic fit of the form y=(FRET$_{max}$)x/($K_d$ +x) (*Hou et al., 2008*) (*Figure 1—figure supplement 1A*). FRET$_{max}$ is defined as the maximal FRET and represents the intrinsic FRET of the bound complex, while $K_d$ represents the protein concentration at which half-maximal FRET is achieved [apparent dissociation constant; in arbitrary units (AU)]. Multiple independent experiments were performed and representative data are shown in *Figure 1—figure supplement 1A*. The mean PLN-SERCA2a FRET$_{max}$ value was 29.9 ± 2.1%, which is similar to previous results (*Hou et al., 2008*; *Hou and Robia, 2010*; *Kelly et al., 2008*), while the FRET$_{max}$ value for DWORF-SERCA2a was 16.3 ± 1.7% (*Figure 1—figure supplement 1B*). This difference in FRET$_{max}$ values is consistent with an increased FRET distance for the DWORF-SERCA2a complex compared to PLN-SERCA2a due to the shorter cytoplasmic domain of DWORF (*Bidwell, 2012*). Importantly, SERCA2a exhibited a higher apparent affinity for DWORF than for PLN as evidenced by a reduction in $K_d$ (*Figure 1A*).

Additionally, we performed progressive acceptor photobleaching experiments to determine the stoichiometry of the SERCA regulatory complexes with PLN or DWORF. We observed a linear increase in donor fluorescence with decreasing acceptor fluorescence, consistent with a 1:1 stoichiometry of the PLN:SERCA and DWORF:SERCA complexes (*Figure 1B* and *Figure 1—figure supplement 1C,D*). PLN has been well described to exist as both a monomer, which is a potent inhibitor of SERCA, and a less inhibitory pentamer (*Kimura et al., 1998*). The modulation of the PLN monomer/pentamer ratio is an important determinant of SERCA activity and therefore cardiac contractility. We performed additional progressive acceptor photobleaching experiments to determine if DWORF is capable of homo-oligomerizing with itself or hetero-oligomerizing with PLN. These experiments did not detect DWORF-DWORF FRET (*Figure 1C*) or PLN-DWORF FRET (*Figure 1D*), suggesting that DWORF does not form homo- or hetero-oligomers at the concentrations achieved here, while PLN-

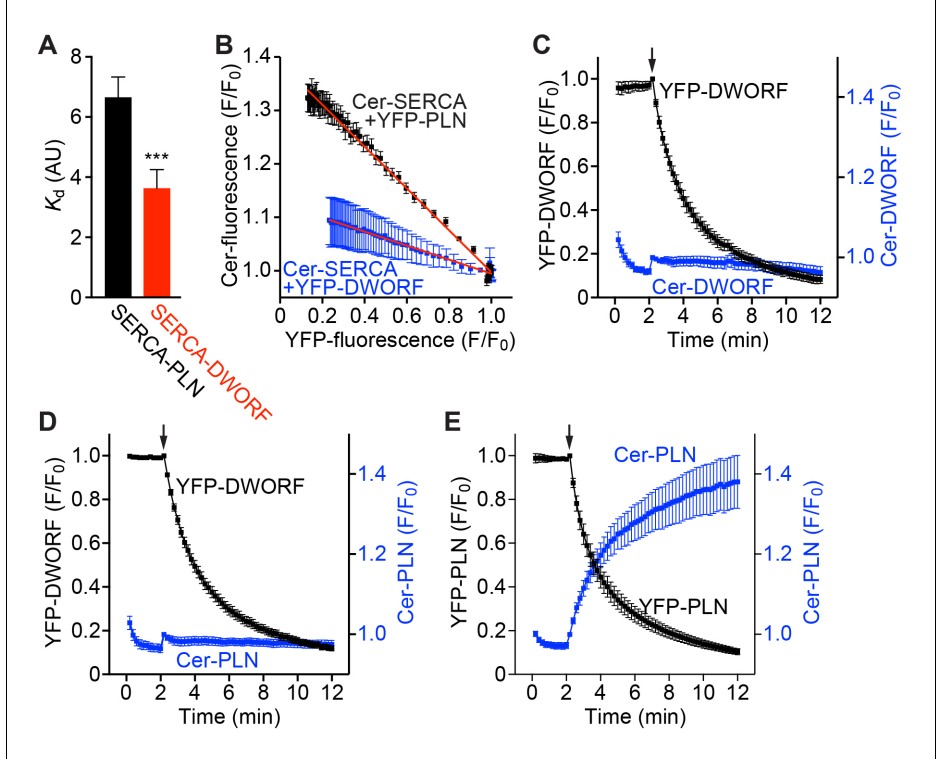

**Figure 1.** FRET analysis indicates that SERCA has a higher apparent affinity for DWORF than PLN and binds both peptides in a 1:1 stoichiometry. (**A**) The apparent affinity of SERCA for PLN and DWORF was assessed by FRET from Cer-SERCA2a to YFP-PLN or YFP-DWORF (Cer excitation, YFP emission). Data are expressed as mean $K_d$ values ± SD for $n = 4$ independent experiments with ~1000 cells analyzed per experiment. A lower $K_d$ value indicates a higher apparent affinity for SERCA. Statistical comparisons between groups were evaluated by Student's t-test. p-value ***p<0.005. AU, arbitrary units. (**B**) Progressive acceptor sensitive photobleaching of YFP-PLN (black) and YFP-DWORF (blue) results in a linear donor vs. acceptor relationship with Cer-SERCA indicating that the stoichiometry of the regulatory complex is 1:1 for both peptides. (**C–E**) Homo-oligomerization of DWORF with itself (**C**) or hetero-oligomerization with PLN (**D**) was not detected by acceptor photobleaching experiments, while PLN-PLN FRET (**E**) showed the expected high-order oligomerization that has been previously described. These data indicate that DWORF exists as a monomer. Progressive acceptor photobleaching experiments (**B–E**) are plotted as mean ± SE. F/F$_0$, fluorescence intensity ratio.

DOI: https://doi.org/10.7554/eLife.38319.003

The following figure supplement is available for figure 1:

**Figure supplement 1.** FRET-based analysis of the interaction of SERCA with PLN and DWORF.

DOI: https://doi.org/10.7554/eLife.38319.004

---

PLN FRET experiments showed the expected high-order oligomerization that has been previously described (*Figure 1E*) (*Kelly et al., 2008*; *Robia et al., 2007*). These results suggest that DWORF exists as a monomer that is available for interaction with SERCA at all times.

## DWORF overexpression prevents impaired Ca²⁺ cycling in PLN transgenic mice

We previously generated DWORF transgenic (Tg) mice using the α-myosin heavy chain (αMHC) promoter to overexpress DWORF specifically in the heart (*Nelson et al., 2016*). Cardiomyocytes from DWORF Tg mice have a cellular phenotype that mimics that observed in PLN null mice, including an increase in peak Ca²⁺ transient amplitude, faster cytosolic Ca²⁺ decay rates, higher SR Ca²⁺ load and enhanced cardiomyocyte contractility (*Nelson et al., 2016*; *Luo et al., 1994*). Our previous work indicates that DWORF activates SERCA by displacing its negative regulator, PLN, and suggests that the profile of enhanced contractility in DWORF Tg animals is due to the ability of DWORF to compete PLN off of SERCA and relieve its inhibitory effects. To investigate this in vivo, we crossed

our DWORF Tg mice with the well-characterized αMHC-PLN transgenic mice (PLN Tg) (*Kadambi et al., 1996*) to generate double transgenic (PLN/DWORF Tg) animals. Cardiomyocytes from PLN Tg animals exhibit a cellular phenotype opposite that of DWORF Tg mice, with reduced peak $Ca^{2+}$ transient amplitude, slower transient decay rates, and reduced fractional shortening due to super-inhibition of SERCA (*Kadambi et al., 1996*). We hypothesized that overexpression of DWORF in PLN Tg mice would lead to displacement of the excess PLN from SERCA and relieve its inhibitory effects.

Baseline cardiac phenotyping of wild-type (WT), PLN Tg, DWORF Tg, or PLN/DWORF Tg mice by echocardiography (ECHO) indicated that all genotypes had similar cardiac function as measured by ejection fraction or fractional shortening (*Figure 2—figure supplement 1A,B*). Additionally, all genotypes analyzed had comparable cardiac dimensions as assessed by ECHO (*Figure 2—figure supplement 1C,D*), normal heart weight to tibia length measurements (*Figure 2—figure supplement 1E*), and similar histological appearances (*Figure 2—figure supplement 1F*), indicating that overexpression of PLN, DWORF, or a combination of the two did not lead to adverse remodeling.

To analyze the cellular phenotype of these animals, we isolated cardiomyocytes from WT, PLN Tg, DWORF Tg, and PLN/DWORF Tg mice and performed $Ca^{2+}$ transient measurements while simultaneously monitoring sarcomere shortening. Consistent with previous findings (*Nelson et al., 2016*), we found that DWORF Tg animals had enhanced $Ca^{2+}$ cycling with increased peak $Ca^{2+}$ transient amplitude and faster transient decay rates (*Figure 2A–C*) accompanied by increased fractional shortening (*Figure 2D,E*). Measurements from PLN Tg cardiomyocytes also recapitulated previous findings and displayed the opposite phenotype characterized by diminished peak $Ca^{2+}$ transient amplitude, slower decay rates and reduced fractional shortening, indicating a strong inhibition of SERCA activity translating into reduced cardiomyocyte contractility (*Figure 2A–E*) (*Kadambi et al., 1996*). Remarkably, cardiomyocytes isolated from PLN/DWORF Tg animals exhibited a complete prevention of impaired $Ca^{2+}$ cycling associated with PLN overexpression (*Figure 2A–E*). PLN/DWORF Tg mice displayed a profile of enhanced $Ca^{2+}$-handling almost identical to that of DWORF Tg animals, indicating that DWORF overexpression can relieve the super-inhibition of SERCA caused by overexpression of PLN.

To directly assess SERCA enzymatic activity in cardiac homogenates from WT, PLN Tg, DWORF Tg and PLN/DWORF Tg mice, we performed oxalate-supported $Ca^{2+}$-dependent $Ca^{2+}$-uptake measurements (*Nelson et al., 2016*; *Bidwell and Kranias, 2016*). Consistent with previously published reports, hearts over-expressing PLN showed a reduction in SERCA activity at lower concentrations of $Ca^{2+}$ substrate quantified as a lower affinity of SERCA for $Ca^{2+}$ (an increase in $K_{Ca}$) (*Figure 2F,G*) (*Nelson et al., 2016*; *Kadambi et al., 1996*), while DWORF Tg hearts exhibited the opposite phenotype with a significant increase in the affinity of SERCA for $Ca^{2+}$ as indicated by a decrease in $K_{Ca}$ (*Figure 2F,G*). SERCA activity assays performed in homogenates from PLN/DWORF Tg mice mirrored those of DWORF Tg mice, indicating that the super-inhibition of SERCA caused by PLN overexpression can be completely nullified in the presence of excess DWORF. Importantly, western blot analysis and quantitative RT-PCR performed on cardiac tissue from WT, PLN Tg, DWORF Tg, and PLN/DWORF Tg mice showed no significant differences in protein or RNA expression levels of any of the major $Ca^{2+}$-handling proteins, indicating that the results observed were not due to compensatory responses (*Figure 2—figure supplement 2A–D*). We also analyzed the phosphorylation state of PLN to verify that our observations were not due to post-translational modifications of the protein that are known to strongly regulate its ability to inhibit SERCA and saw no significant changes amongst genotypes (*Figure 2—figure supplement 2B and D*) (*Luo et al., 1998*). Taken together, these results support previous data indicating that DWORF overexpression enhances cardiac $Ca^{2+}$ cycling and contractility through displacement of PLN from SERCA, thereby relieving its inhibitory effects (*Nelson et al., 2016*). To further substantiate these findings, we analyzed the interaction of SERCA2a with PLN and DWORF in a heterologous expression system. HEK293 cells were co-transfected with equal amounts of Myc-tagged SERCA2a and HA-tagged PLN in the presence of increasing levels of HA-DWORF, and Myc-SERCA2a/HA-peptide interactions were assessed by Myc (SERCA2a) immunoprecipitation and western blot analysis. We observed a strong reduction in the interaction of HA-PLN with SERCA2a when co-expressed with HA-DWORF, and this occurred in a dose-dependent manner (*Figure 2—figure supplement 3A*). Consistent with previous findings (*Nelson et al., 2016*), using the same heterologous expression system we found that co-expression of DWORF with SERCA2a did not change the apparent affinity of SERCA for $Ca^{2+}$, but it relieved

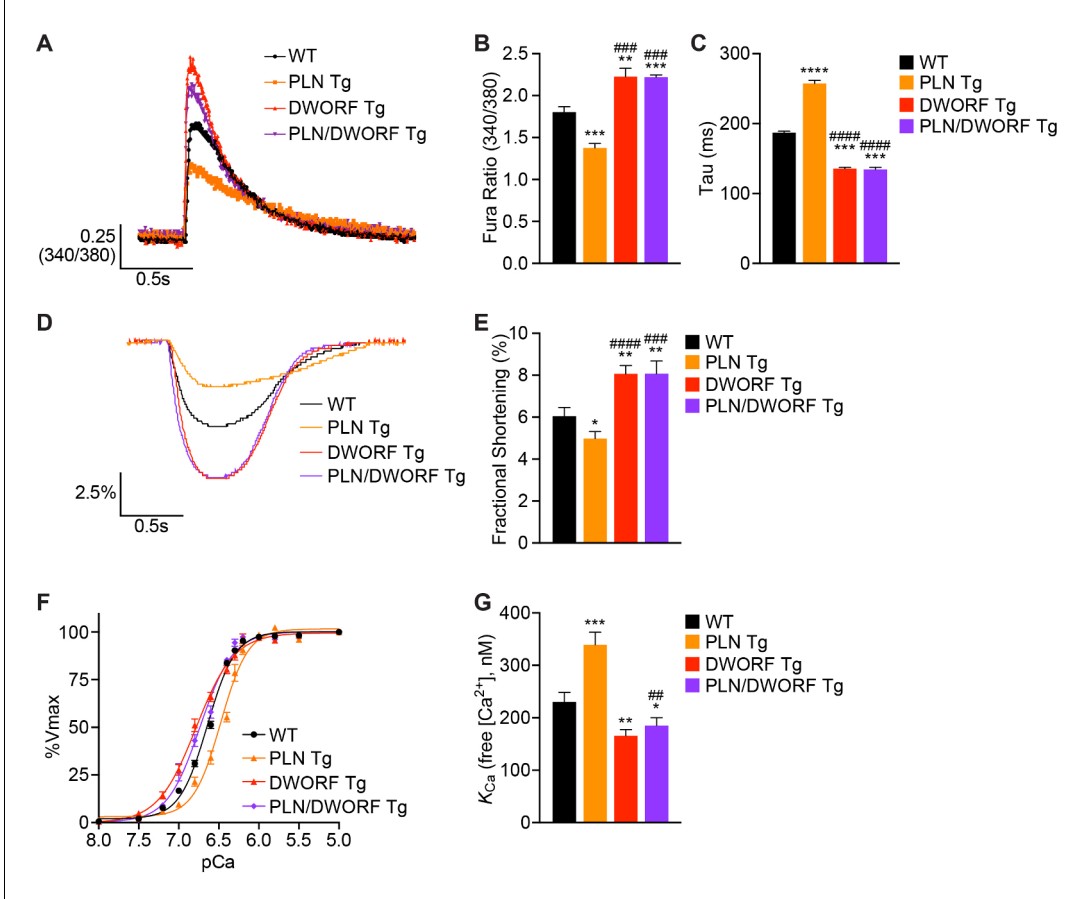

**Figure 2.** DWORF overexpression counteracts the inhibitory effects of excess PLN on SERCA in vivo. (**A**) Representative $Ca^{2+}$ transients recorded in Fura-2 loaded cardiomyocytes from WT, PLN Tg, DWORF Tg, and PLN/DWORF Tg mice. (**B**) Mean peak amplitude of pacing-induced $Ca^{2+}$ transients and transient decay rates (tau) (**C**) in Fura-2 loaded cardiomyocytes from WT, PLN Tg, DWORF Tg, and PLN/DWORF Tg mice. Transient decay rates were measured by fitting a single exponential to the decay phase of the $Ca^{2+}$ transient. (**D**) Representative fractional shortening tracings as measured by sarcomere length during cardiomyocyte contraction. (**E**) Mean fractional shortening data from mice with the indicated genotypes. Data are represented as mean ±SD for $n = 3$ animals with 10–12 recordings per animal. Statistical comparisons between groups were evaluated by Student's t-test. p-value *$p<0.05$, **$p<0.01$, ***$p<0.005$ or ****$p<0.001$ vs. WT and ###$p<0.005$ or ####$p<0.001$ vs. PLN Tg. (**F, G**) $Ca^{2+}$-dependent $Ca^{2+}$-uptake assays were performed using total homogenates from hearts of WT, PLN Tg, DWORF Tg, and PLN/DWORF Tg mice to directly measure SERCA affinity for $Ca^{2+}$ ($K_{Ca}$) and SERCA activity. Representative tracings (**F**) and average $K_{Ca}$ values (**G**) from $n = 4$ hearts of each genotype are represented as bar graphs (±SD). *P*-value *$p<0.05$, **$p<0.01$ or ***$p<0.005$ vs. WT and ##$p<0.01$ vs. PLN Tg.

DOI: https://doi.org/10.7554/eLife.38319.005

The following figure supplements are available for figure 2:

**Figure supplement 1.** Cardiac function and histological analysis of PLN/DWORF Tg mice.

DOI: https://doi.org/10.7554/eLife.38319.006

**Figure supplement 2.** Analysis of RNA and protein expression levels of the major cardiac $Ca^{2+}$-handling proteins in PLN/DWORF Tg mice.

DOI: https://doi.org/10.7554/eLife.38319.007

**Figure supplement 3.** DWORF binding to SERCA2a displaces PLN in a dose-dependent manner and enhances SERCA activity.

DOI: https://doi.org/10.7554/eLife.38319.008

the inhibition of PLN on SERCA in a dose-dependent manner (*Figure 2—figure supplement 3B,C*). These results substantiate the hypothesis that the overexpression of DWORF could be a powerful means of enhancing SERCA activity via the displacement of PLN and therefore may enhance cardiac contractility in the setting of heart failure and prevent the progression of the disease.

## DWORF overexpression prevents cardiac dysfunction in MLP KO mice

To directly assess the potential of DWORF as a therapeutic for heart failure, we crossed our DWORF Tg mice with the well-characterized MLP KO mouse model of DCM. The MLP protein is expressed in cardiac and skeletal muscle and is predominantly localized adjacent to the Z-disc where it plays a structural role and also acts as a stress signaling molecule that transduces mechanical stress into biochemical signals (*Arber et al., 1997*; *Arber et al., 1994*; *Heineke et al., 2005*; *Knöll et al., 2002*). The adult-onset DCM phenotype exhibited by MLP KO mice mimics that of human DCM and is characterized by progressive dilation of all four cardiac chambers, ventricular wall thinning, a reduction in cardiac contractility and elongation of action potential duration (*Arber et al., 1997*; *Hoshijima et al., 2006*). Notably, defects in SR $Ca^{2+}$ cycling have been shown to be important determinants of cardiac dysfunction and the transition to heart failure in MLP KO mice (*Minamisawa et al., 1999*). We have previously shown that DWORF mRNA and protein levels are dramatically reduced in human ischemic heart failure and in mouse models of cardiovascular disease, indicating that a decrease in DWORF expression may contribute to the $Ca^{2+}$ dysregulation that drives cardiac decompensation (*Nelson et al., 2016*). We measured DWORF expression in cardiac tissue from WT and MLP KO mice and found a reduction in both protein and RNA levels in MLP KO hearts (*Figure 3A,B*), suggesting that loss of DWORF expression may contribute to the DCM phenotype.

To evaluate whether DWORF overexpression provides cardioprotection in MLP KO mice, we crossed our DWORF Tg animals with MLP KO mice to create MLP KO/DWORF Tg mice. Cardiac function was assessed in 8-week-old mice by ECHO (*Figure 3C*). Consistent with previous reports, MLP KO mice showed a marked reduction in left ventricular (LV) function compared to WT animals as measured by ejection fraction (*Figure 3D*) and fractional shortening (*Figure 3E*). Cardiac-specific overexpression of DWORF in MLP KO mice resulted in a significant improvement of LV function as evidenced by increases in ejection fraction and fractional shortening to values approaching those of WT mice (*Figure 3C–E*). We also assessed whether DWORF loss-of-function exacerbated the MLP KO phenotype and indeed observed that MLP/DWORF double KO (dKO) mice showed a further decline in cardiac function as compared to MLP KO mice (*Figure 3C–E*). Cardiac dimensions were calculated from M-mode ECHO tracings and MLP KO mice showed an increase in LV internal diameter both during diastole, or relaxation (*Figure 3F*), and systole, or contraction (*Figure 3G*), consistent with the clinical presentation of DCM. Concurrent loss of DWORF protein in MLP/DWORF dKO mice resulted in a slight but non-significant increase in chamber dilation, while DWORF overexpression led to a dramatic reduction in LV chamber dilation and near complete prevention of the MLP KO DCM phenotype (*Figure 3F,G*).

Diastolic dysfunction coexists in human patients with dilated cardiomyopathy, and it has previously been shown that the progression to heart failure in MLP KO mice may be anticipated by diastolic cardiac dysfunction (*Lorenzen-Schmidt et al., 2005*). We evaluated LV diastolic function in our mice by pulse-wave Doppler echocardiography of transmitral valve blood flow and by mitral annular tissue Doppler (*Table 1*). We found that the E/A ratio (ratio of the early [E] to late [A] ventricular filling velocities, *Figure 3H*) and E/E' ratio (ratio of early filling [E] to early diastolic mitral annular velocity [E'], *Figure 3I*) of MLP KO and MLP/DWORF dKO were significantly greater than WT animals. In MLP KO/DWORF Tg mice, both the E/A ratio (*Figure 3H*) and E/E' ratio (*Figure 3I*) were indistinguishable from those of WT mice, indicating that DWORF overexpression ameliorates the diastolic dysfunction observed in MLP KO mice (*Figure 3H and I* and *Table 1*). Collectively, these results indicate that the restoration of SERCA activity and enhancement of $Ca^{2+}$ cycling in MLP KO mice via DWORF overexpression is sufficient to prevent the onset of DCM in MLP KO mice and their subsequent transition to heart failure.

## DWORF overexpression prevents pathological remodeling in MLP KO mice

Histological analysis of 8-week-old MLP KO hearts showed characteristic morphological defects consistent with DCM including ventricular and atrial chamber dilation, wall thinning and cardiac enlargement, and these features were exacerbated in MLP/DWORF dKO mice (*Figure 4A*). In sharp contrast, overexpression of DWORF in MLP KO mice prevented the spectrum of morphological defects observed in MLP KO hearts (*Figure 4A*). Significant ventricular cardiomyocyte hypertrophy

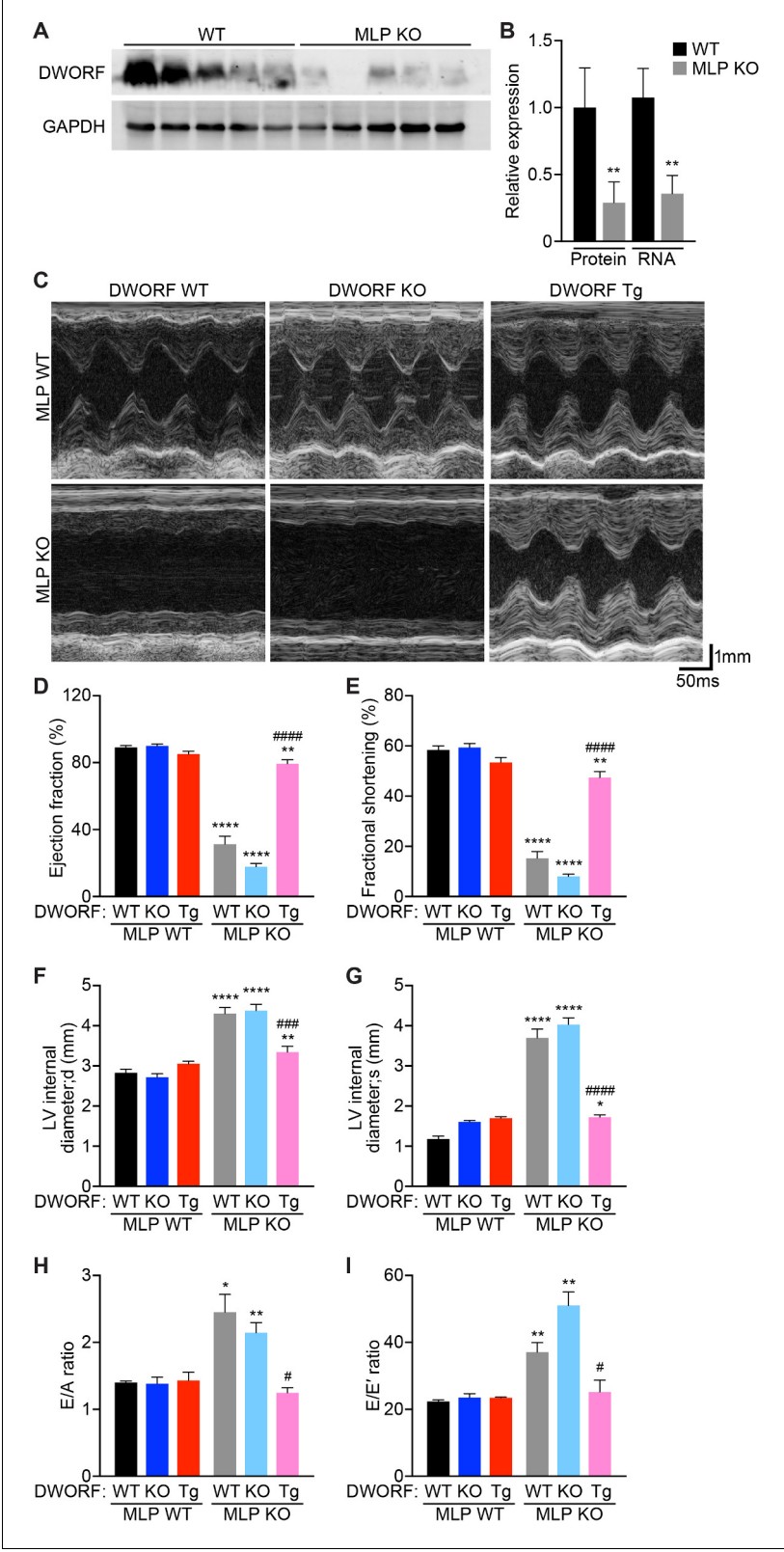

**Figure 3.** Overexpression of DWORF ameliorates cardiac dysfunction in MLP KO mice. (**A**) Western blot analysis of heart homogenates from 8-week-old WT and MLP KO mice immunoblotted with anti-DWORF antibody. (**B**) Quantification of expression of DWORF protein (left) and RNA (right) in 8-week-old WT and MLP KO hearts. Data are expressed as mean ±SD for *n* = 5 animals. p-value **p<0.01 vs. WT. (**C**) Representative M-mode
*Figure 3 continued on next page*

*Figure 3 continued*

echocardiographic tracings from 8-week-old mice with the indicated genotypes. Cardiac function was evaluated from M-mode images and is represented by mean percent ejection fraction (**D**) and percent fractional shortening (**E**). Cardiac dimensions were also measured and are represented as left ventricular (LV) internal diameter during maximal relaxation (diastole, d) (**F**) and contraction (systole, s) (**G**). Data are expressed as mean ±SD for $n$ = 5–15 mice per genotype. p-value *p<0.05, **p<0.01, or ****p<0.001 vs. WT and ###p<0.005 or ####p<0.001 vs. MLP KO. (**H, I**) Indices of diastolic function were assessed by Doppler echocardiography and are quantified and represented as E/A ratio (**H**) and E/E' ratio (**I**). Data were collected from $n$ = 3–5 mice per genotype and are expressed as mean ±SD. p-value *p<0.05 or **p<0.01 vs. WT and #p<0.05 vs. MLP KO. Statistical comparisons between groups were evaluated by Student's t-test.

DOI: https://doi.org/10.7554/eLife.38319.009

was observed in both MLP KO and MLP/DWORF dKO mice compared to WT animals as assessed by cross-sectional area analysis (*Figure 4—figure supplement 1A,B*) and isolated cardiomyocyte length and width measurements (*Figure 4—figure supplement 1C–E*). DWORF overexpression in MLP KO mice significantly blunted this hypertrophic response and cell size parameters were indistinguishable from WT mice in the MLP KO/DWORF Tg group (*Figure 4—figure supplement 1A–E*). Additionally, MLP KO/DWORF Tg hearts had heart weight to tibia length (*Figure 4B*) and lung weight to tibia length (*Figure 4C*) measurements comparable to those of WT mice, while MLP KO and MLP/DWORF dKO mice showed significant increases in these parameters, indicative of advanced heart failure. MLP/DWORF dKO mice also had a significantly higher liver weight to tibia length ratio compared to any of the other genotypes assessed, indicating that these animals were in a particularly aggravated state of congestive heart failure (*Figure 4D*).

Quantification of cardiac fibrosis by Picrosirius Red staining revealed significant myocardial fibrosis in MLP KO mice that was mildly exacerbated in MLP/DWORF dKO mice and dramatically reduced in MLP KO/DWORF Tg mice at 8 weeks of age (*Figure 5A,B*). Quantitative RT-PCR revealed a robust induction of the cardiac fetal gene program in MLP KO mice, a molecular marker of pathological cardiac hypertrophy (*Figure 5C*). This response was significantly inhibited in MLP KO/DWORF Tg mice, which is consistent with the preservation of ventricular function in these animals (*Figure 5C*). Ultrastructural analysis of MLP KO mice by electron microscopy revealed a striking disruption of cardiac myofibrillar organization characteristic of the late phases of DCM in both mice and humans (*Figure 5D*). Overexpressing DWORF in MLP KO mice resulted in complete prevention of these ultrastructural defects, indicating a preservation of cardiac function and cardiomyocyte architecture (*Figure 5D*).

**Table 1.** Echo-Doppler assessment of left ventricular diastolic function in the different experimental groups of mice.
Data are represented as mean ±SD for $n$ = 3–5 mice per genotype. p-value *p<0.05, **p<0.01 or ***p<0.005 vs. WT and #p<0.05 vs. MLP KO. Statistical comparisons between groups were evaluated by Student's t-test. Abbreviation used: E, peak Doppler blood inflow velocity across mitral valve during early diastole; A, peak Doppler blood inflow velocity across mitral valve during late diastole; E', peak tissue Doppler of myocardial relaxation velocity at mitral valve annulus during early diastole; A', peak tissue Doppler of myocardial relaxation velocity at mitral valve annulus during late diastole; HR, heart rate; bpm, beats per minute.

| Genotype | E mm/s | A mm/s | E' mm/s | A' mm/s | E/A | E/E' | HR bpm |
|---|---|---|---|---|---|---|---|
| WT | 569.7 ± 39.0 | 408.0 ± 39.2 | 25.6 ± 2.4 | 10.7 ± 1.8 | 1.4 ± 0.1 | 22.3 ± 0.8 | 465.7 ± 22.0 |
| DWORF KO | 591.5 ± 22.1 | 432.1 ± 61.8 | 25.2 ± 2.0 | 19.3 ± 1.4* | 1.4 ± 0.2 | 23.5 ± 2.0 | 447.3 ± 8.3 |
| DWORF Tg | 624.2 ± 43.2 | 443.6 ± 81.8 | 26.6 ± 1.7 | 11.8 ± 1.3 | 1.4 ± 0.2 | 23.4 ± 0.4 | 445.7 ± 5.0 |
| MLP KO | 546.0 ± 38.7 | 236.9 ± 78.1* | 15.1 ± 2.8** | 14.7 ± 2.3* | 2.4 ± 0.6* | 37.1 ± 6.3** | 438.6 ± 19.3 |
| MLP/DWORF dKO | 536.9 ± 60.2 | 251.2 ± 24.6** | 10.6 ± 1.0***, # | 13.6 ± 1.0 | 2.1 ± 0.3** | 51.0 ± 7.0** | 444.3 ± 21.7 |
| MLP KO/DWORF Tg | 606.4 ± 199.6 | 492 ± 182.7# | 24.0 ± 4.1# | 18.7 ± 3.1 | 1.2 ± 0.1# | 25.2 ± 6.1* | 467 ± 14.1 |

DOI: https://doi.org/10.7554/eLife.38319.010

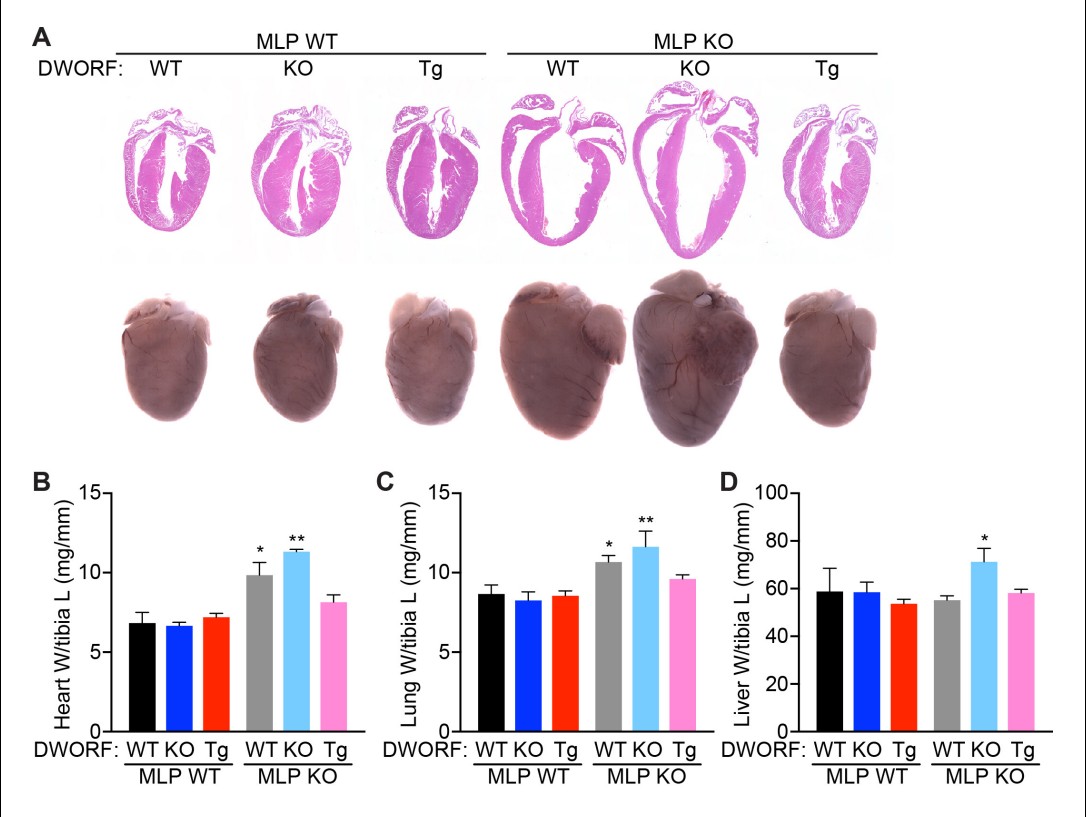

**Figure 4.** DWORF overexpression abrogates the DCM phenotype of MLP KO mice. (**A**) Representative hematoxylin and eosin (H and E) staining of four-chamber histological sections (top) or whole mount hearts (bottom) from 8-week-old mice with the indicated genotypes. (**B**) Heart weight (W) to tibia length (L), lung weight to tibia length (**C**), and liver weight to tibia length (**D**) measurements from $n$ = 3–7 mice per genotype. Data are represented as mean ±SD. Statistical comparisons between groups were evaluated by Student's t-test. p-value *$p < 0.05$, **$p < 0.01$ vs. WT.

DOI: https://doi.org/10.7554/eLife.38319.011

The following figure supplement is available for figure 4:

**Figure supplement 1.** Overexpression of DWORF in MLP KO mice prevents cardiomyocyte hypertrophy.
DOI: https://doi.org/10.7554/eLife.38319.012

## DWORF overexpression in MLP KO mice enhances Ca²⁺ cycling and myocyte contractility

To gain further insight into the mechanisms responsible for the dramatic improvement of cardiac function in MLP KO mice by DWORF overexpression, we isolated cardiomyocytes from our animals and performed intracellular Ca²⁺ transients and fractional shortening measurements. Compared to WT mice, MLP KO and MLP/DWORF dKO cardiomyocytes exhibited marked reductions in Ca²⁺ transient amplitude (*Figure 6A*), significant prolongation of the transient decay rate (*Figure 6B*), and decreased fractional shortening (*Figure 6C*), collectively indicating diminished SERCA activity and Ca²⁺ cycling. Additionally, sarcomere relaxation kinetics were significantly slowed in MLP KO and MLP/DWORF dKO cardiomyocytes (*Figure 6D*). Overexpression of DWORF in MLP KO mice resulted in an increase in cardiomyocyte Ca²⁺ transient amplitude (*Figure 6A*), faster transient decay rates (*Figure 6B*), enhanced fractional shortening (*Figure 6C*) and increased sarcomere relaxation kinetics (*Figure 6D*) to levels that prevented the phenotype observed in MLP KO mice and surpassed those of WT cardiomyocytes. Sarcomere shortening kinetics were similar across all genotypes analyzed (*Figure 6E*), indicating a specific alteration in cardiomyocyte relaxation kinetics in this animal model. We directly confirmed that SERCA enzymatic activity was enhanced in MLP KO/DWORF Tg animals by performing oxalate supported Ca²⁺-dependent Ca²⁺-uptake measurements in cardiac

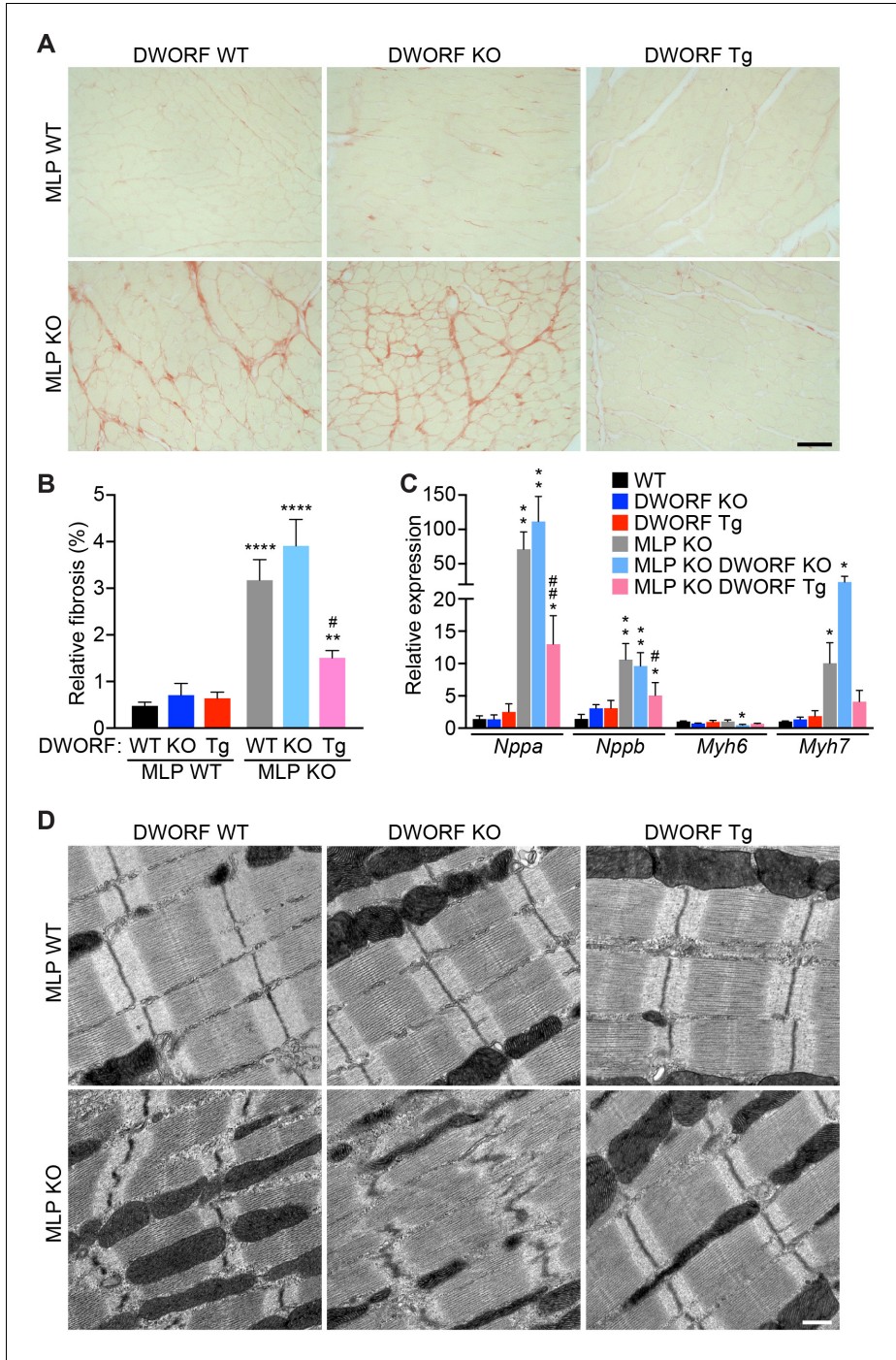

**Figure 5.** Overexpression of DWORF in MLP KO mice prevents fibrosis and ultrastructural defects and mitigates the hypertrophic gene response. (**A**) Representative Picrosirius red staining of heart sections from animals with the specified genotypes. The red color indicates fibrosis. Scale bar: 50 µm. (**B**) Quantification of Picrosirius red staining in *n* = 5–9 mice. Data are expressed as the mean fibrotic area relative to the total myocardium (±SD). Statistical comparisons between groups were evaluated by Student's t-test. p-value **p<0.01 or ****p<0.001 vs. WT and #p<0.05 vs. MLP KO. (**C**) qRT-PCR of 8-week-old hearts from MLP KO mice and MLP KO/DWORF KO mice show a robust activation of the fetal gene program that is typical of the pathological hypertrophic response, which is diminished with DWORF overexpression. *Nppa*, atrial natriuretic peptide; *Nppb*, brain natriuretic peptide; *Myh6*,α-myosin heavy chain; *Myh7*, β-myosin heavy chain. Data are normalized to 18S values and are presented as expression level relative to WT, mean ±SD for *n* = 4–5 mice per genotype. p-value *p<0.05 or **p<0.01 vs. WT and #p<0.05 or ##p<0.01 vs. MLP KO. (**D**) Consistent with the characteristics of DCM, electron micrographs of heart

*Figure 5 continued on next page*

*Figure 5 continued*

sections from MLP KO mice exhibit pronounced myofibrillar disarray which is prevented with DWORF overexpression. Images shown are representative of $n$ = 3 mice per genotype. Scale bar: 0.5 μm.
DOI: https://doi.org/10.7554/eLife.38319.013

homogenates and observed a strong leftward shift of the SERCA activity curve (*Figure 6F*), indicating an increase in the affinity of SERCA for $Ca^{2+}$ and quantified as a reduction in $K_{Ca}$ value (*Figure 6G*).

Consistent with previous reports (*Minamisawa et al., 1999*), western blotting and quantitative RT-PCR revealed that MLP gene deletion does not cause significant alterations in protein or RNA levels of any major $Ca^{2+}$ handling genes in the heart, suggesting that the defects of $Ca^{2+}$ cycling in MLP KO mice result from a functional impairment of excitation-contraction coupling rather than a decrease in the proteins mediating the cycling itself (*Figure 6—figure supplement 1*). We also analyzed the phosphorylation state and oligomerization of PLN to verify that our observations were not due to post-translational modifications of the protein that are known to strongly regulate its ability to inhibit SERCA and saw no significant changes amongst genotypes (*Figure 6—figure supplement 1A and C*). Taken together, these findings provide evidence that the reversal of the MLP KO phenotype by DWORF overexpression mechanistically lies in the ability of DWORF to displace PLN from SERCA and enhance its activity to restore $Ca^{2+}$ cycling and maintain cardiac contractility.

## Discussion

While heart failure is a complex disease with many distinctly different causes, the functional characteristics of the failing myocardium are surprisingly consistent and include the slowing of both contraction and relaxation rates and the prolongation of the cardiac action potential (*Houser et al., 2000*). Alterations in $Ca^{2+}$ cycling and depressed SR $Ca^{2+}$ re-uptake are universal features of heart failure that have been shown to contribute directly to the pathogenesis of cardiovascular disease (*Piacentino et al., 2003*; *Luo and Anderson, 2013*). For this reason, significant attention has been focused on restoring $Ca^{2+}$ homeostasis through enhancing SERCA activity, which has been shown to maintain cardiac contractility and prevent the progression of the disease (*Gwathmey et al., 2013*; *Kranias and Hajjar, 2012*; *Pleger et al., 2013*; *Ly et al., 2007*). Here we describe a novel approach to stimulate SERCA activity through DWORF overexpression and present strong evidence demonstrating its potent ability both to enhance $Ca^{2+}$ cycling and contractility and to prevent the development of cardiomyopathy in a well-characterized mouse model of DCM.

In the heart, it is well established that SERCA activity is reversibly regulated by PLN, a small transmembrane protein that directly interacts with SERCA and reduces its activity by lowering its affinity for $Ca^{2+}$. Since its discovery over 40 years ago, the regulation of PLN and its interaction with SERCA have been the subject of intense research (*Kranias and Hajjar, 2012*; *MacLennan and Kranias, 2003*; *Nelson et al., 2014*; *Vangheluwe et al., 2006*; *Hou et al., 2008*; *Hou and Robia, 2010*; *Kelly et al., 2008*; *Kimura et al., 1998*; *Robia et al., 2007*). Our lab recently identified DWORF as a novel transmembrane protein that resides in the cardiac SR membrane and competes for the same binding site on SERCA as PLN and enhances SERCA activity (*Nelson et al., 2016*). This work has opened up new avenues of research aimed at understanding SERCA regulation in the heart and also provides a novel mechanism to increase SERCA activity in the context of heart failure. In this study, we used FRET to investigate the stoichiometry and relative affinity of micropeptide regulatory complexes in live cell membranes and found that SERCA has a higher apparent affinity for DWORF than PLN, which makes it an attractive candidate for gene therapy. While enhancing SERCA activity by increasing its expression level has been a primary focus of gene therapy studies thus far, there is also strong experimental evidence that increasing SERCA activity through the ablation of the SERCA inhibitor PLN is beneficial (*Minamisawa et al., 1999*; *Sato et al., 2001*), and achieving a similar outcome through DWORF overexpression is a much more clinically relevant approach to achieve this goal.

In this study, we used the MLP KO mouse model of DCM to test our hypothesis that increasing SERCA function by DWORF overexpression would prevent the development of ventricular dysfunction, fibrosis, and long-term heart failure that is characteristic of dilated cardiomyopathy. We

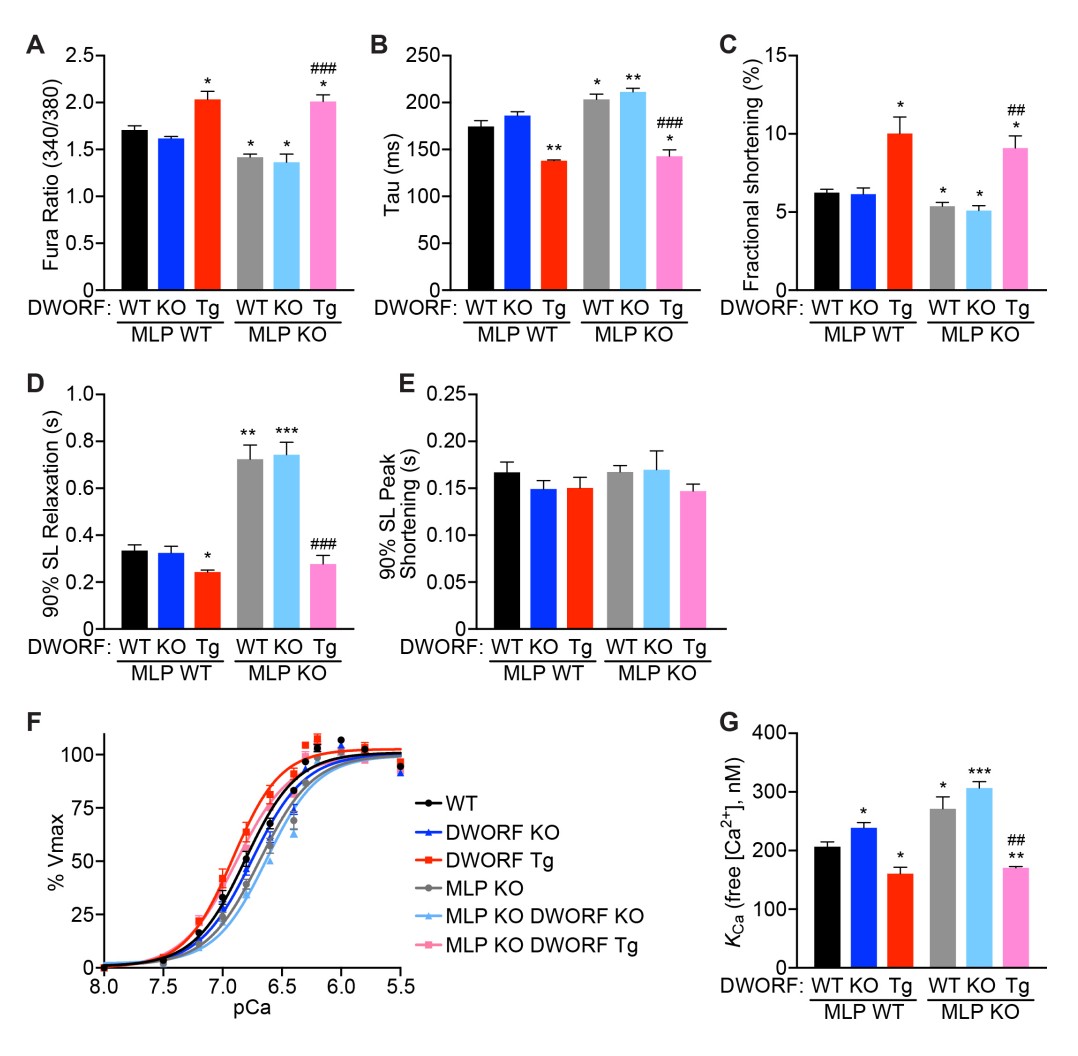

**Figure 6.** DWORF overexpression enhances $Ca^{2+}$ cycling, contractility and SERCA activity in MLP KO mice. (**A**) Average pacing-induced peak $Ca^{2+}$ transient amplitude in isolated adult cardiomyocytes loaded with Fura-2 from 8-week-old animals with the indicated genotypes. (**B**) Mean $Ca^{2+}$ transient decay rates (tau) were calculated by fitting a single exponential to the decay phase of the $Ca^{2+}$ transient. (**C**) Average peak cardiomyocyte fractional shortening analysis as measured by sarcomere length during contraction from mice with the indicated genotypes. (**D, E**) Sarcomere relaxation (**D**) and contraction (**E**) kinetics were analyzed from sarcomere length measurements during pacing-induced contractions (SL, sarcomere length). Data are represented as mean ±SD for *n* = 4 animals with 6–12 recordings per animal. Statistical comparisons between groups were evaluated by Student's t-test. p-value *p<0.05 or **p<0.01 vs. WT and ##p<0.01 or ###p<0.005 vs. MLP KO. (**F, G**) Oxalate supported $Ca^{2+}$-dependent $Ca^{2+}$-uptake assays were performed using total homogenates from hearts of mice with the indicated genotypes to directly measure SERCA affinity for $Ca^{2+}$ ($K_{Ca}$) and SERCA activity. Representative tracings (**F**) and average $K_{Ca}$ values (**G**) from *n* = 4 hearts of each genotype are presented as mean ±SD. p-value *p<0.05, **p<0.01 or ***p<0.005 vs. WT and ##p<0.01 vs. MLP KO.

DOI: https://doi.org/10.7554/eLife.38319.014

The following figure supplement is available for figure 6:

**Figure supplement 1.** Expression and post-translational modifications of $Ca^{2+}$-handling proteins.

DOI: https://doi.org/10.7554/eLife.38319.015

selected this specific genetic model of DCM for several reasons. First, genetic ablation of MLP in mice has been shown to lead to adult onset DCM, which closely resembles the human disease, and MLP KO mice represent a very common and extensively used mouse model for studying the patho-physiology of dilated cardiomyopathy (*Arber et al., 1997*; *Minamisawa et al., 1999*; *Lorenzen-*

*Schmidt et al., 2005*; *Heineke et al., 2010*; *Recchia and Lionetti, 2007*; *Rockman et al., 1998*). Second, it has been shown that MLP KO mice exhibit the characteristic cardiomyocyte $Ca^{2+}$ cycling defects that are acquired in human heart failure, including reductions in peak cardiomyocyte $Ca^{2+}$ transient amplitude and prolongation of $Ca^{2+}$ reuptake kinetics, which relates to decreased SERCA activity (*Arber et al., 1997*), and these are the specific parameters that should be enhanced with DWORF overexpression. Lastly, it has been shown that increasing SERCA function by PLN gene deletion in MLP KO mice preserved $Ca^{2+}$ homeostasis and prevented the development of DCM in this model (*Minamisawa et al., 1999*). Because we believe that DWORF exerts its effect through the displacement of PLN to induce maximal cardiomyocyte contractility, we expected to see a similar degree of benefit in MLP KO mice via DWORF overexpression. Interestingly, it has been reported that in ischemic cardiomyopathy and idiopathic DCM, MLP protein levels are significantly decreased, implying that MLP deficiency may be a common pathophysiological mechanism in advanced heart failure (*Zolk et al., 2000*). Our studies largely focused on $Ca^{2+}$ dynamics due to the described function of DWORF as a potent stimulator of SERCA activity, but it should be noted that SERCA is not the only target that could rescue this animal model of DCM. In fact, it has previously been shown that G-protein-coupled receptor kinase 2 (GRK2) inhibition via overexpression of a β-adrenergic receptor kinase peptide inhibitor (βARKct) prevented the development of myocardial failure in MLP KO mice (*Rockman et al., 1998*). We did not directly assess β-adrenergic receptor density or responsiveness in our animals, so this could also be a contributing factor. Additionally, we showed that DWORF overexpression in MLP KO mice resulted in a dramatic attenuation of fibrosis compared to MLP KO mice, which also may contribute to its ability to prevent the disease phenotype. It remains to be seen whether DWORF overexpression can preserve cardiac function in other forms of DCM and in chronic heart failure. In the future, we will test the therapeutic potential of DWORF overexpression in additional clinically relevant models of heart failure by adeno-associated viral (AAV) delivery. It has previously been shown that enhancing contractility by overexpressing SERCA is protective against diabetic cardiomyopathy (*Trost et al., 2002*) as well as cardiac dysfunction induced by chronic pressure overload (*Miyamoto et al., 2000*; *del Monte et al., 2001*; *Nakayama et al., 2003*) and we believe that we can achieve the same results with DWORF overexpression.

Despite promising results in rodent and large animal models of heart failure (*Gwathmey et al., 2013*; *Kawase et al., 2008*; *Miyamoto et al., 2000*; *Prunier et al., 2008*; *Ly et al., 2007*), overexpression of SERCA by AAV gene delivery in a human clinical trial (Calcium Up-regulation by Percutaneous administration of gene therapy In cardiac Disease, CUPID) failed to meet its primary endpoints and was discontinued (*Greenberg et al., 2016*; *Greenberg et al., 2014*). A major reason this clinical trial is believed to have failed was due to lack of successful SERCA overexpression (*Greenberg et al., 2016*). In spite of the failure of this clinical trial, we believe that gene therapy is still a promising approach for heart failure treatment and that DWORF overexpression may be superior to that of SERCA. First, as mentioned above, the CUPID trial likely failed primarily because SERCA, which is a large multi-pass transmembrane protein, was not efficiently delivered to the targeted cells and expressed in the cells it did infect. In this regard, DWORF may provide a more optimal protein for delivery due to its small size, which allows it to be easily packaged in AAV vectors and rapidly translated from a relatively small number of transcripts. Second, in heart failure, it is well established that SERCA levels are reduced, but there is often either no change or a slight increase in PLN expression which would dramatically increase the PLN to SERCA ratio (*Kranias and Hajjar, 2012*). Therefore, overexpressing SERCA alone may not be sufficient to overcome this imbalance. Since DWORF can enhance the activity of SERCA in the presence of excess PLN (*Figure 2*), it may prove to be more beneficial to increase the activity of the endogenous SERCA pump by expressing DWORF rather than the pump itself. Lastly, it has been shown that SERCA2a requires post-translational modification with SUMO for full activity (*Kho et al., 2011*), a process that may be limited by the capacity for SUMOylation rather than SERCA abundance. Ectopic expression of DWORF could increase the activity of the available SERCA protein without the need to address SUMOylation capacity. We strongly believe that enhancing $Ca^{2+}$ cycling remains a compelling pathway to target in the development of heart failure therapeutics as its disruption is a major common insult in the disease and our evidence suggests that the overexpression of DWORF represents a potent means to achieve this goal.

In summary, through multiple independent assays, our results show that DWORF displays a higher apparent affinity for SERCA than PLN, partly because PLN has a high self-affinity for oligomerization. Thus, the more monomeric DWORF outcompetes PLN for binding to the pump. Overexpression of DWORF leads to the displacement of PLN from SERCA and results in enhanced SERCA activity, even in instances where PLN is overexpressed causing SERCA super-inhibition. Lastly, the prevention of the DCM phenotype of MLP KO mice by DWORF overexpression highlights the clinical potential of DWORF overexpression as a promising therapeutic for heart failure and an attractive candidate for future gene therapy studies.

# Materials and methods

## Key resources table

| Reagent type | Designation | Source | Identifiers | Additional information |
|---|---|---|---|---|
| Antibody | Mouse Anti-HA Tag Monoclonal Antibody (5B1D10) | Invitrogen | Cat. #32–6700 | WB (1:2,000) |
| Antibody | Mouse Anti-Myc Tag Monoclonal Antibody | Invitrogen | Cat. #R950-25 | WB (1:2,000), IP (1 ug) |
| Antibody | Mouse Anti-Phospholamban Monoclonal Antibody (2D12) | Invitrogen | Cat. #MA3-922 | WB (1:2,000) |
| Antibody | Rabbit Anti-Phospholamban (PLN, PLB) (pSer16) pAb | Badrilla | Cat. #A010-12AP | WB (1:1,000) |
| Antibody | Rabbit Anti-Phospholamban (PLN, PLB) (pThr17) pAb | Badrilla | Cat. #A010-13AP | WB (1:1,000) |
| Antibody | Mouse Anti-Ryanodine Receptor Monoclonal Antibody (C3-33) | Invitrogen | Cat. #MA3-916 | WB (1:1,000) |
| Antibody | Mouse Anti-SERCA2 ATPase Monoclonal Antibody (2A7-A1) | Invitrogen | Cat. #MA3-919 | WB (1:1,000) |
| Antibody | Rabbit Anti-Calsequestrin Polyclonal Antibody | Invitrogen | Cat. #PA1-913 | WB (1:1,000) |
| Antibody | Rabbit Anti-DWORF | New England Peptide | Custom made | WB (1:1,000) |
| Antibody | Mouse Anti-GAPDH Loading Control Monoclonal Antibody (GA1R) | Invitrogen | Cat. #MA5-15738 | WB (1:10,000) |
| Antibody | Rabbit Anti-Calcium Channel Antibody, Voltage Gated $\alpha$1C, pAb | Millipore | Cat. #AB5156 | WB (1:250) |
| Antibody | Goat Anti-Mouse IgG (H + L) -HRP Conjugate | Bio-Rad | Cat. #1706516 | WB (1:20,000) |
| Antibody | Goat Anti-Rabbit IgG (H + L) -HRP Conjugate | Bio-Rad | Cat. #1706515 | WB (1:20,000) |
| Antibody | DyLight 800 Sheep Anti-Rabbit IgG | Bio-Rad | Cat. #STAR36D800GA | WB (1:10,000) |
| Antibody | StarBright Blue 700 Goat Anti-Mouse IgG | Bio-Rad | Cat. #12004159 | WB (1:10,000) |
| Antibody | hFAB Rhodamine Anti-GAPDH Primary Antibody | Bio-Rad | Cat. #12004168 | WB (1:10,000) |
| Antibody | Mouse Anti-Cardiac Troponin T Monoclonal Antibody [1C11] | Abcam | Cat. #ab8295 | IHC (1:500) |
| Antibody | Goat anti-Mouse IgG (H + L) Cross-Adsorbed Secondary Antibody, Alexa Fluor 555 | Invitrogen | Cat. #A-21422 | IHC (1:500) |

*Continued on next page*

*Continued*

| Reagent type | Designation | Source | Identifiers | Additional information |
|---|---|---|---|---|
| Other | Wheat Germ Agglutinin, Alexa Fluor 488 Conjugate | Invitrogen | Cat. #W11261 | IHC (1:500) |
| Other | VECTASHIELD Antifade Mounting Medium with DAPI | Vector Laboratories | Cat. #H-1200 | IHC |
| Other | Dynabeads Protein G for Immunoprecipitation | Invitrogen | Cat. #10004D | IP |

## Experimental design

The objectives of the present study were to molecularly characterize the interaction of DWORF with SERCA and to directly test if DWORF overexpression could prevent the development of cardiomyopathy in a mouse model of DCM. Male mice were used for all experiments and all mice with the appropriate genotypes were used without any exclusions. With the exception of echocardiography measurements, we did not use blinding approaches. All echocardiography experiments were performed and analyzed by a single blinded operator. The sample sizes were based on previous experience and published reports. For each experiment, sample size is indicated in the figure legend and reflects the number of independent biological replicates. In general, sample size was chosen to use the least number of animals to achieve statistical significance, and no statistical methods were used to predetermine sample size.

## FRET measurements

AAV-293 cells were cultured in DMEM cell culture medium supplemented with 10% fetal bovine serum (FBS) (ThermoScientific, Waltham, MA) and transiently transfected using MBS mammalian transfection kit (Agilent Technologies, Stratagene, La Jolla, CA), according to manufacturer instructions. The transfected cells were trypsinized (ThermoScientific) and replated onto poly D lysine-coated glass-bottom chambers and allowed to adhere for 1–2 hr prior to imaging. Acceptor sensitization FRET microscopy was performed as described previously (*Hou et al., 2008*; *Hou and Robia, 2010*). Cells were imaged with an inverted microscope (Nikon Eclipse Ti) equipped with an EM-CCD camera (iXon 887, Andor Technology, Belfast, Northern Ireland). Acquisition was performed with a $40 \times 0.75$ N.A. objective with 100 ms exposure for each channel: Cer, YFP, and 'FRET' (Cer excitation, YFP emission). Fluorescence intensity was quantified from ~1000 cells per sample using automatic multiwavelength cell scoring in MetaMorph (Molecular Devices, Sunnyvale, CA). FRET efficiency was calculated according to $E = G/(G + 3.2 \times F_{Cer})$, where $G = F_{FRET} - a \times F_{YFP} - d \times F_{Cer}$ (*Himes et al., 2016*), where $F_{FRET}$, $F_{YFP}$, and $F_{Cer}$ are the matching fluorescence intensity from FRET, YFP, and Cer images, respectively, and $G$ represents FRET intensity corrected for the bleed-through of the channels. The parameters $a$ and $d$ are bleed-through constants calculated as $a = F_{FRET}/F_{Cer}$ for a control sample transfected with only YFP-SERCA and $d = F_{FRET}/F_{Cer}$ for a control sample transfected with only Cer-SERCA. These values were determined to be $G = 4.74$ $a = 0.075$ and $b = 0.88$. Progressive acceptor photobleaching was performed as described previously (*Kelly et al., 2008*; *Zak et al., 2017*). Briefly, we collected images of Cer and YFP fluorescence at intervals to establish a baseline and then initiated progressive acceptor photobleaching (*Zak et al., 2017*), acquiring successive images of Cer and YFP in between 10 s of exposure to illumination through a 504/12 nm bandpass filter for selective photobleaching of YFP. The images were analyzed in FIJI (*Schindelin et al., 2012*), and FRET was calculated from the pre- and post-bleach donor fluorescence intensity using the equation $FRET = 1 - (F_{DA}/F_D)$ where $F_{DA}$ = the intensity of the donor before bleaching and $F_D$ = the intensity of the donor after bleaching. To distinguish between 1:1 and higher order stoichiometry, the fluorescence of the donor was plotted against the fluorescence of the acceptor at the same time point during progressive bleaching. A linear relationship was taken to indicate a 1:1 complex of Cer- and YFP-labeled proteins (*Kelly et al., 2008*; *Zak et al., 2017*).

## Mice

Animal work described in this manuscript has been approved and conducted under the oversight of the UT Southwestern Institutional Animal Care and Use Committee. Mice were housed in a barrier facility with a 12 hr light/dark cycle and maintained on standard chow (2916 Teklad Global, Houston, TX). All mouse lines used in this manuscript have been previously published (*Arber et al., 1997*; *Nelson et al., 2016*; *Kadambi et al., 1996*). All data presented were collected from male mice.

## Transthoracic echocardiography (ECHO)

Cardiac function and heart dimensions were determined by two-dimensional echocardiography using a Visual Sonics Vevo 2100 Ultrasound (Visual Sonics,Toronto, Canada) on conscious mice. M-mode tracings were used to measure anterior and posterior wall thicknesses at end diastole and end systole. Left ventricular (LV) internal diameter (LVID) was measured as the largest anteroposterior diameter in either diastole (LVID;d) or systole (LVID;s). A single observer blinded to mouse genotypes performed echocardiography and data analysis. Fractional shortening (FS) was calculated according to the following formula: FS(%) = [(LVID;d − LVID;s)/LVID;d]×100. Ejection fraction (EF%) was calculated by: EF(%)=([EDV − ESV]/EDV)×100. EDV, end diastolic volume; ESV, end systolic volume. Diastolic function was assessed in lightly anesthetized mice (1.5%–2% isoflurane) using pulsed wave Doppler recordings of the maximal early (E) and late (A) diastolic transmitral flow velocities and Doppler tissue imaging recordings of peak E' velocity and peak A' velocity in apical four-chamber view. Body temperature was maintained at 37°C throughout using a heating pad. Changes in transmitral flow pattern (E/A ratio) and mitral annulus velocities (E', A') were used to assess diastolic dysfunction.

## Adult mouse cardiomyocyte isolation

Adult mouse hearts were rapidly excised and the aorta was cannulated on a constant-flow Langendorff perfusion apparatus. Hearts were digested with perfused Tyrode's solution (10 mM glucose, 5 mM HEPES, 5.4 mM KCl, 1.2 mM $MgCl_2$, 150 mM NaCl, 2 mM sodium pyruvate, pH 7.4) containing Liberase (0.25 mg/ml), and the ventricles were minced, filtered, and equilibrated with Tyrode's solution containing 1 mM $CaCl_2$ and bovine serum albumin at room temperature (*Nelson et al., 2016*). Cardiomyocyte length and width measurements were assessed using ImageJ analysis on bright-field images taken of freshly isolated cardiomyocytes imaged with a 20X objective. Length/width measurements were taken at the longest/widest part of each cell.

## Cardiomyocyte $Ca^{2+}$ transients and contractility measurements

Adult cardiomyocytes were loaded with 0.5 μM Fura-2-AM (Molecular Probes, Eugene, OR) and placed in a heated chamber (37°C) on the stage of an inverted microscope. The chamber was perfused with Tyrode's solution containing $CaCl_2$ (1.8 mM) (pH 7.4). Cardiomyocytes were paced with an IonOptix Myocyte Calcium and Contractility System at 0.5 Hz using a MyoPacer field stimulator. Changes in intracellular $Ca^{2+}$ levels were monitored using Fura-2 dual-excitation (340/380 nm), single emission (510 nm) ratiometric imaging. Tau, the decay rate of the average $Ca^{2+}$ transient trace, was determined using IonWizard 6.0 analysis software (IonOptix, Westwood, MA). Cardiomyocyte contractility measurements were made using sarcomere length (SarcLen) parameters and data was processed with IonWizard 6.0 analysis software.

## Oxalate-supported $Ca^{2+}$ uptake measurements

Oxalate-supported $Ca^{2+}$ uptake in cardiac homogenates and transfected HEK293 cells were measured as previously described in detail (*Nelson et al., 2016*; *Bidwell and Kranias, 2016*). Briefly, mouse hearts were isolated and rapidly snap frozen in liquid nitrogen and stored at −80°C until processed. Frozen tissue samples or cultured cells were homogenized in 50 mM phosphate buffer, pH 7.0 containing 10 mM NaF, 1 mM EDTA, 0.3 M sucrose, 0.3 mM PMSF and 0.5 mM DTT. $Ca^{2+}$ uptake was measured in reaction solution containing 40 mM imidazole pH 7.0, 95 mM KCl, 5 mM $NaN_3$, 5 mM $MgCl_2$, 0.5 mM EGTA, 5 mM $K^+$ oxalate, 1 μM ruthenium red and various concentrations of $CaCl_2$ to yield 0.02 to 5 μM free $Ca^{2+}$. The reaction was initiated by the addition of ATP (final concentration 5 mM). The data were analyzed by nonlinear regression with computer software

(GraphPad Software), and the $K_{Ca}$ values were calculated using an equation for a general cooperative model for substrate activation.

## Co-immunoprecipitations (CoIPs)

CoIPs were performed as previously described (*Nelson et al., 2016*). Briefly, HEK293 cells were co-transfected with expression plasmids encoding Myc-SERCA2a and HA-PLN in the presence of increasing concentrations of HA-DWORF or control plasmid. Whole cell lysates were prepared in CoIP buffer (20 mM $NaPO_4$, 150 mM NaCl, 2 mM $MgCl_2$, 0.1% NP-40, 10% Glycerol, 10 mM sodium fluoride, 0.1 mM sodium orthovanadate, 10 mM sodium pyrophosphate, 1 mM DTT and Complete protease inhibitor [Roche, Basel, Switzerland]). Immunoprecipitations were carried out using 1 mg of mouse monoclonal anti-Myc antibody (Invitrogen, Carlsbad, CA) and collected with Dynabeads (Invitrogen, Carlsbad, CA). Tris/Tricine gel electrophoresis was performed using pre-cast 16.5% Mini-PROTEAN Tris-Tricine gels (Bio-Rad, Hercules, CA). Standard western blot procedures were performed on input and IP fractions using the following antibodies: HA (Invitrogen, Carlsbad, CA), Myc (Invitrogen, Carlsbad, CA) or GAPDH (Invitrogen, Carlsbad, CA). Specific catalogue numbers for the reagents used for these CoIP studies can be found in the Key Resource Table.

## Histology and immunofluorescence

Hearts were isolated and fixed in 4% (vol/vol) paraformaldehyde in PBS for 48 hr at 4°C with gentle shaking. Hearts were dehydrated, embedded in paraffin, and sectioned. Heart sections were stained with hematoxylin and eosin (H and E) and Picrosirius red using standard procedures. Fibrosis was quantified using Pircorsirius red staining and ImageJ software (NIH, Rockville, MD). For immunofluorescent staining, tissue sections were deparaffinized and subjected to antigen retrieval with Citra buffer (BioGenex, Fremont, CA). Tissue sections were incubated with 488-conjugated Wheat Germ Agglutinin (Invitrogen, Carlsbad, CA) to label the cell membranes and cardiomyocytes were immunostained using a primary antibody for cardiac troponin-T (Abcam, Cambridge, MA) and an Alexa Fluor 555 secondary antibody (Invitrogen, Carlsbad, CA). Coverslips were mounted using VECTA-SHIELD Antifade Mounting Media with DAPI (Vector Laboratories, Burlingame, CA) and confocal images were taken of the mid-LV free wall with a Zeiss LSM-800 using a 40X oil objective. Cardiomyocyte cross-sectional area was assessed using Fiji Software. Specific catalogue numbers for the reagents used for immunohistochemistry can be found in the Key Resource Table.

## Transmission electron microscopy

Eight-week-old mice were perfusion fixed by transcardial perfusion using 4% paraformaldehyde and 1% glutaraldehyde in 0.1 M sodium cacodylate buffer (pH 7.4). Heart tissue was collected and samples were processed by the University of Texas Southwestern Medical Center Electron Microscopy Core facility. Briefly, fixed tissues were post-fixed, stained, dehydrated, and embedded in EMbed-812 resin. Tissue sections were cut and post-stained, and images were acquired on a FEI Tecnai G2 Spirit TEM.

## Quantitative mRNA measurement

Total RNA was extracted from adult mouse tissues using Trizol and reverse transcribed using iScript Reverse Transcription Supermix (Bio-Rad, Hercules, CA) with random primers. Quantitative Polymerase Chain Reaction (qPCR) reactions were assembled using KAPA Probe Fast qPCR Master Mix (SIGMA, St. Louis, MO) and the following TaqMan probes from Applied Biosystems (Foster City, CA): *Atp2a2* (Mm01201431_m1), *Ryr2* (Mm00465877_m1), *Cacna1c* (Mm01188822_m1), *Casq2* (Mm00486742_m1), *Pln* (Mm00452263_m1), *Nppa* (Mm01255747_g1), *Nppb* (Mm01255770_g1), *Myh6* (Mm00440359_m1) and *Myh7* (Mm01318999_g1). Assays were performed using a 7900HT Fast Real-Time PCR machine (Applied Biosystems). Expression was normalized to *18S* mRNA using Kappa SYBR Fast qPCR Master Mix and was represented as fold change relative to wild-type. *18S* and *DWORF* (currently annotated as *Gm34302*) oligonucleotides were ordered from Integrated DNA Technologies:

 *18* s Forward: 5'- ACC GCA GCT AGG AAT AAT GGA −3'
 *18* s Reverse: 5'- GCC TCA GTT CCG AAA ACC A −3'
 *DWORF* Forward: 5'- TTC TTC TCC TGG TTG GAT GG −3'

*DWORF* Reverse: 5′- TCT TCT AAA TGG TGT CAG ATT GAA GT −3′

## Tissue western blot analysis

Tissues were collected and snap frozen in liquid nitrogen. Frozen samples were pulverized and homogenized in RIPA buffer (SIGMA) with added cOmplete, EDTA-free protease inhibitor cocktail (Roche, Basel, Switzerland) and PhosSTOP phosphatase inhibitors (Roche) on ice. Protein concentration was determined using a Pierce BCA Protein Assay Kit (ThermoFisher Scientific, Waltham, MA). Samples were separated on Mini-PROTEAN TGX Precast Gels (Bio-Rad, Hercules, CA) or bis/acrylamide gels made by standard gel preparation. Gels were transferred to PVDF membrane (Millipore, Immobilon-P, Burlington, MA), blocked in 5% milk/TBST and then incubated in primary antibodies: total PLN (2D12, Invitrogen, Carlsbad, CA), pSer16-PLN and pThr17-PLN (Badrilla, Leeds, UK); SERCA2 (2A7-A1, Invitrogen, Carlsbad, CA); RyR2 (Invitrogen, Carlsbad, CA, C3-33); LTCC (α1C, Millipore, Burlington, MA); Calsequestrin (Invitrogen, Carlsbad, CA); DWORF (custom antibody, New England Peptide, Gardner, MA) (*Nelson et al., 2016*); GAPDH (Invitrogen, Carlsbad, CA). Western blots were washed in TBST, incubated with fluorescent or HRP-conjugated secondary antibodies (Bio-Rad, Hercules, CA), and then developed using a ChemiDoc MP Imagine System (Bio-Rad, Hercules, CA) or autoradiograph film. Westerns were quantified using ImageJ software (NIH) using an internal GAPDH loading control for each western blot analyzed. Specific catalogue numbers for the antibodies used for westerns can be found in the Key Resource Table.

## Statistical information

All statistical analyses were performed using Prism 6 (GraphPad, San Diego, CA). Information on the statistical analyses presented are included in each figure legend and are either mean ±SD or SEM. Two-tailed t-tests were performed to determine significance. p-Values were defined as follows: *, #p<0.05, **, ##p<0.01, ***, ###p<0.005 or ****, ####p<0.001. All samples were included.

## Acknowledgements

We thank E Kranias (University of Cincinnati College of Medicine) for generously providing the phospholamban transgenic mice. We thank J Cabrera for graphics, W Tan for echocardiography, the Molecular Pathology Core under the direction of J Richardson at the University of Texas Southwestern Medical Center for help with histology, and the Electron Microscopy Core at the University of Texas Southwestern Medical Center under the direction of K Luby-Phelps for help with EM sample processing. This work was supported by grants from the NIH (HL-130253, HD-087351, HL-138426, HL-092321 and AR-067294), Fondation Leducq Networks of Excellence, and the Robert A Welch Foundation (grant 1–0025 to ENO). CAM was supported by a National Heart, Lung, and Blood Institute, NIH Ruth L Kirschstein National Research Service Award (NRSA) (F32HL129674) and a National Heart, Lung, and Blood Institute, NIH Pathway to Independence Award (K99HL141630). GGS was supported by an American Heart Association and the Theodore and Beulah Beasley Foundation grant (18POST34060230). Funding sources were not involved in study design, data collection and interpretation, or the decision to submit the work for publication.

## Additional information

### Competing interests

Catherine A Makarewich: CAM has filed a provisional patent application to use DWORF for treatment of heart failure (Application #62/324,706). Rhonda Bassel-Duby: RB-D has filed a provisional patent application to use DWORF for treatment of heart failure (Application #62/324,706). Eric N Olson: ENO has filed a provisional patent application to use DWORF for treatment of heart failure (Application #62/324,706). The other authors declare that no competing interests exist.

### Funding

| Funder | Grant reference number | Author |
| --- | --- | --- |
| National Institutes of Health | F32 (HL-129674) | Catherine A Makarewich |

| National Institutes of Health | K99 (HL-141630) | Catherine A Makarewich |
|---|---|---|
| American Heart Association | 18POST34060230 | Gabriele G Schiattarella |
| National Institutes of Health | R01 (HL-092321) | Seth L Robia |
| National Institutes of Health | R01 (HL-130253) | Rhonda Bassel-Duby<br>Eric N Olson |
| National Institutes of Health | R01 (HL-138426) | Rhonda Bassel-Duby |
| Welch Foundation | 1–0025 | Eric N Olson |
| Fondation Leducq | | Eric N Olson |
| National Institutes of Health | R01 (HD-087351) | Eric N Olson |
| National Institutes of Health | R01 (AR-067294) | Eric N Olson |

The funders had no role in study design, data collection and interpretation, or the decision to submit the work for publication.

## Author contributions

Catherine A Makarewich, Conceptualization, Data curation, Formal analysis, Funding acquisition, Validation, Investigation, Visualization, Methodology, Writing—original draft, Writing—review and editing; Amir Z Munir, Conceptualization, Data curation, Formal analysis, Validation, Investigation, Writing—review and editing; Gabriele G Schiattarella, Conceptualization, Data curation, Formal analysis, Methodology, Writing—review and editing; Svetlana Bezprozvannaya, Data curation, Validation, Investigation; Olga N Raguimova, Ellen E Cho, Data curation, Formal analysis, Investigation, Methodology; Alexander H Vidal, Data curation, Formal analysis; Seth L Robia, Conceptualization, Data curation, Formal analysis, Supervision, Funding acquisition, Validation, Investigation, Visualization, Methodology, Project administration, Writing—review and editing; Rhonda Bassel-Duby, Supervision, Funding acquisition, Project administration, Writing—review and editing; Eric N Olson, Conceptualization, Supervision, Funding acquisition, Visualization, Writing—original draft, Project administration, Writing—review and editing

## Author ORCIDs

Catherine A Makarewich https://orcid.org/0000-0001-6907-4401
Eric N Olson https://orcid.org/0000-0003-1151-8262

## Ethics

Animal experimentation: This study was performed in strict accordance with the recommendations in the Guide for the Care and Use of Laboratory Animals of the National Institutes of Health. Animal work described in this manuscript has been approved and conducted under the oversight of the UT Southwestern Institutional Animal Care and Use Committee (IACUC). All of the animals were handled according to approved IACUC protocols (#2017-102269 and #2016-101833) of the UT Southwestern Medical Center.

## Decision letter and Author response

Decision letter https://doi.org/10.7554/eLife.38319.018
Author response https://doi.org/10.7554/eLife.38319.019

## Additional files

### Supplementary files

• Transparent reporting form
DOI: https://doi.org/10.7554/eLife.38319.016

### Data availability

All data generated or analysed during this study are included in the manuscript and supporting files.

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
