## [Decision Letter]

Thank you for submitting your article "The DWORF Micropeptide Enhances Contractility and Prevents Heart Failure in a Mouse Model of Dilated Cardiomyopathy" for consideration by *eLife*. Your article has been reviewed by Didier Stainier as the Senior Editor, a Reviewing Editor, and three reviewers. The reviewers have opted to remain anonymous. They were enthusiastic about your article, but also raised some concerns.

The reviewers have discussed the reviews with one another and the Reviewing Editor has drafted this decision to help you prepare a revised submission.

Summary:

This manuscript reports preclinical evaluation of DWORF (DWarf Open Reading Frame) peptide as a potent activator of SERCA in the failing heart, and an interesting alternative to SERCA gene therapy. DWORF is shown to be a competitive displacer of PLN at SERCA2a with higher affinity than PLN for SERCA. Cardiac directed transgenic DWORF peptide expression antagonized the hallmark cardiac myocyte phenotypes of transgenic PLN overexpression and prevented characteristic calcium cycling abnormalities and dilated cardiomyopathy in MLP deficient mice. These highly detailed and rigorous experiments markedly extend the findings from the seminal report in Science and make a case for further evaluation of DWORF therapy in heart failure. The biophysical studies are especially nice. The PLN-DWORF transgenic cross studies had the predicted outcome based on the underlying mechanistic model, but that in no way compromises the need to report such studies.

Essential revisions:

1) One limitation of the work is the exclusive reliance on genetic models for cardiac disease. The MLP knockout mouse is a model of dilated cardiomyopathy, but it is not typical of clinical heart failure, and there are more clinically relevant models. Also, micropeptide delivery to the heart in the DWORF transgenic mouse before the disease manifested is not directly clinically translatable. The authors should explain their rationale for experimental design and discuss the limitations of the MLP KO model.

2) The MLP knockout is a unique genetic model. Please discuss if DORF would elicit inotropic response in other chronic heart failure models.

3) The authors should address the additional data needed to demonstrate that the altered regulation is merely due to a competition between PLN and DWORF. Some of the possibilities include:

a) Adding a group with unregulated SERCA {plus minus} DWORF to show that there is no direct/additive effect of DWORF on SERCA to pump function.

b) A competition assay between DWORF and PLN performed on the PLN KO background (using virus), to show that the mechanism is loss of PLN regulation.

c) Immunoprecipitate SERCA to show at which concentrations only DWORF and not PLN gets precipitated

4) The blot in Figure 2—figure supplement 2 shows large variability in expression. Please explain whether enough samples were examined, and whether the sample variability was caused by technical reasons.

5) As there is a sizeable increase in Ca^2+^ amplitude and change in Ca^2+^ decay kinetics in the DWORF Tg and DWORF-MLP KO, are sarcomere relaxation kinetics also faster in these animals?

---

## [Author Response]

Essential revisions:1) One limitation of the work is the exclusive reliance on genetic models for cardiac disease. The MLP knockout mouse is a model of dilated cardiomyopathy, but it is not typical of clinical heart failure, and there are more clinically relevant models. Also, micropeptide delivery to the heart in the DWORF transgenic mouse before the disease manifested is not directly clinically translatable. The authors should explain their rationale for experimental design and discuss the limitations of the MLP KO model.

We agree with the reviewers comments and have modified the Discussion section of our manuscript to address these points. In brief, the MLP KO mouse model has been extensively used for studying the pathophysiology of dilated cardiomyopathy because it leads to adult onset DCM, which closely resembles the human disease. It has been shown that MLP KO mice develop the characteristic Ca^2+^ cycling defects that are classically acquired in human heart failure including reductions in peak Ca^2+^ transient amplitude and prolongation of Ca^2+^ reuptake kinetics. Therefore, this model provides an excellent platform to test if enhancing SERCA activity via DWORF overexpression can rescue cardiac function and ameliorate their heart failure phenotype. We do recognize that there are limitations to our experimental design and we have edited the Discussion section of our manuscript to acknowledge these.

2) The MLP knockout is a unique genetic model. Please discuss if DORF would elicit inotropic response in other chronic heart failure models.

We understand the reviewers’ point and have edited the Discussion section of our manuscript to address this. We do believe that DWORF overexpression will enhance contractility and be beneficial in other chronic heart failure models. SERCA overexpression has previously been shown to be protective against several different animal models of heart failure including diabetic cardiomyopathy and pressure-overload induced heart failure. In our future studies, we will assess whether enhancing SERCA activity via DWORF overexpression (by transgene or AAV mediated gene-delivery) can provide similar protection in these chronic heart failure models.

3) The authors should address the additional data needed to demonstrate that the altered regulation is merely due to a competition between PLN and DWORF. Some of the possibilities include:a) Adding a group with unregulated SERCA {plus minus} DWORF to show that there is no direct/additive effect of DWORF on SERCA to pump function.b) A competition assay between DWORF and PLN performed on the PLN KO background (using virus), to show that the mechanism is loss of PLN regulation.c) Immunoprecipitate SERCA to show at which concentrations only DWORF and not PLN gets precipitated

We thank the reviewers for these suggestions. We have performed suggested experiments ‘a’ and ‘c’ and have included these results as Figure 2—figure supplement 3. Consistent with our previous publication on DWORF (Nelson et al., 2016), we found that co-expression of DWORF with unregulated SERCA did not lead to a direct or additive stimulation of SERCA activity (Figure 2—figure supplement 3B and C). Instead, we found that the ability of DWORF to simulate SERCA activity lies in its capacity to compete PLN off of SERCA and reduce the inhibitory effects of PLN on SERCA. This competitive displacement of PLN from SERCA can be very clearly seen in immunoprecipitation experiments shown in Figure 2—figure supplement 3A.

4) The blot in Figure 2—figure supplement 2 shows large variability in expression. Please explain whether enough samples were examined, and whether the sample variability was caused by technical reasons.

We have increased the number of samples analyzed by Western and have included a new panel of representative Western blots alongside the previous panel (Figure 2—figure supplement 2B). While we do still see a small level of variation in protein expression between samples, it does not trend in the direction of significance between groups in any of the proteins analyzed. Several of the calcium-handling proteins we are analyzing, namely RyR2 and the LTCC, are rather large and can be difficult to cleanly detect by Western blotting. We ran each sample multiple times to confirm our results and performed all quantification analyses using internal GAPDH controls for each blot (although only a single GAPDH blot is shown in Figure 2—figure supplement 2B for clarity).

5) As there is a sizeable increase in Ca^2+^ amplitude and change in Ca^2+^ decay kinetics in the DWORF Tg and DWORF-MLP KO, are sarcomere relaxation kinetics also faster in these animals?

We thank the reviewer for this suggestion. We re-analyzed the fractional shortening traces we previously obtained using sarcomere length measurements and found that sarcomere relaxation kinetics are indeed faster in DWORF Tg mice (Figure 6D). Additionally, we found that sarcomere relaxation kinetics are significantly slowed in MLP KO and MLP/DWORF dKO mice and this parameter is normalized with DWORF overexpression (Figure 6D).